# Neural Execution of Graph Algorithms

**Petar Veličković**
DeepMind
petarv@google.com

**Rex Ying**[*]
Stanford University
rexying@stanford.edu

**Matilde Padovano**[*]
University of Cambridge
mp861@cam.ac.uk

**Raia Hadsell**
DeepMind
raia@google.com

**Charles Blundell**
DeepMind
cblundell@google.com

## Abstract

Graph Neural Networks (GNNs) are a powerful representational tool for solving problems on graph-structured inputs. In almost all cases so far, however, they have been applied to directly recovering a final solution from raw inputs, without explicit guidance on how to *structure* their problem-solving. Here, instead, we focus on learning in the space of *algorithms*: we train several state-of-the-art GNN architectures to imitate individual steps of classical graph algorithms, parallel (breadth-first search, Bellman-Ford) as well as sequential (Prim's algorithm). As graph algorithms usually rely on making discrete decisions within neighbourhoods, we hypothesise that maximisation-based message passing neural networks are best-suited for such objectives, and validate this claim empirically. We also demonstrate how learning in the space of algorithms can yield new opportunities for *positive transfer* between tasks—showing how learning a shortest-path algorithm can be substantially improved when simultaneously learning a reachability algorithm.

## 1 Introduction

A multitude of important real-world tasks can be formulated as tasks over graph-structured inputs, such as navigation, web search, protein folding, and game-playing. Theoretical computer science has successfully discovered effective and highly influential algorithms for many of these tasks. But many problems are still considered intractable from this perspective.

Machine learning approaches have been applied to many of these classic tasks, from tasks with known polynomial time algorithms such as *shortest paths* (Graves et al., 2016; Xu et al., 2019) and *sorting* (Reed & De Freitas, 2015), to intractable tasks such as *travelling salesman* (Vinyals et al., 2015; Bello et al., 2016; Kool et al., 2018), *boolean satisfiability* (Selsam et al., 2018; Selsam & Bjørner, 2019), and even *probabilistic inference* (Yoon et al., 2018). Recently, this work often relies on advancements in graph representation learning (Bronstein et al., 2017; Hamilton et al., 2017; Battaglia et al., 2018) with graph neural networks (GNNs) (Li et al., 2015; Kipf & Welling, 2016; Gilmer et al., 2017; Veličković et al., 2018). In almost all cases so far, ground-truth *solutions* are used to drive learning, giving the model complete freedom to find a mapping from raw inputs to such solution[1].

Many classical algorithms share related *subroutines*: for example, shortest path computation (via the Bellman-Ford (Bellman, 1958) algorithm) and breadth-first search both must enumerate sets of edges adjacent to a particular node. Inspired by previous work on the more general tasks of program synthesis and learning to execute (Zaremba & Sutskever, 2014; Kaiser & Sutskever, 2015; Kurach et al., 2015; Reed & De Freitas, 2015; Santoro et al., 2018), we show that by learning several algorithms simultaneously and providing a supervision signal, our neural network is able to demonstrate positive knowledge transfer between learning different algorithms. The supervision signal is driven by how a known classical algorithm would process such inputs (including any relevant *intermediate outputs*), providing explicit (and *reusable*) guidance on how to tackle graph-structured problems. We call this approach **neural graph algorithm execution**.

---

[*]Work performed while the author was at DeepMind.

[1]We note that there exist good reasons for choosing this approach, e.g. ease of optimisation.

Given that the majority of popular algorithms requires making *discrete decisions* over neighbourhoods (e.g. *"which edge should be taken?"*), we suggest that a highly suitable architecture for this task is a message-passing neural network (Gilmer et al., 2017) with a *maximisation* aggregator—a claim we verify, demonstrating clear performance benefits for simultaneously learning breadth-first search for reachability with the Bellman-Ford algorithm for shortest paths. We also verify its applicability to *sequential* reasoning, through learning Prim's algorithm (Prim, 1957) for minimum spanning trees.

Note that our approach complements Reed & De Freitas (2015): we show that a relatively simple graph neural network architecture is able to learn and algorithmically transfer among different tasks, do not require explicitly denoting subroutines, and tackle tasks with superlinear time complexity.

## 2 PROBLEM SETUP

### 2.1 GRAPH ALGORITHMS

We consider graphs $G = (V, E)$ where $V$ is the set of nodes (or vertices) and $E$ is the set of edges (pairs of nodes). We will consider graphs for two purposes: 1) as part of the task to be solved (e.g., the graph provided as input to breadth first search), 2) as the input to a graph neural network.

A graph neural network receives a sequence of $T \in \mathbb{N}$ graph-structured inputs. For each element of the sequence, we will use a fixed $G$ and vary meta-data associated with the nodes and edges of the graph in the input. In particular, to provide a graph $G = (V, E)$ as input to a graph neural network, each node $i \in V$ has associated node features $\vec{x}_i^{(t)} \in \mathbb{R}^{N_x}$ where $t \in \{1, \ldots, T\}$ denotes the index in the input sequence and $N_x$ is the dimensionality of the node features. Similarly, each edge $(i, j) \in E$ has associated edge features $\vec{e}_{ij}^{(t)} \in \mathbb{R}^{N_e}$ where $N_e$ is the dimensionality of the edge features. At each step, the algorithm produces node-level outputs $\vec{y}_i^{(t)} \in \mathbb{R}^{N_y}$. Some of these outputs may then be reused as inputs on the next step; i.e., $\vec{x}_i^{(t+1)}$ may contain some elements of $\vec{y}_i^{(t)}$.

### 2.2 LEARNING TO EXECUTE GRAPH ALGORITHMS

We are interested in learning a graph neural network that can execute one or more of several potential algorithms. The specific algorithm to be executed, denoted $A$, is provided as an input to the network. The structure of the graph neural network follows the *encode-process-decode* paradigm (Hamrick et al., 2018). First we will describe the encode-process-decode architecture and then describe the specific parameterisations that are used for each sub-network. The relation between the variables used by the graph algorithm and our neural algorithm executor setup is further illustrated in Figure 1.

For each algorithm $A$ we define an *encoder network* $f_A$. It is applied to the current input features and previous *latent features* $\vec{h}_i^{(t-1)}$ (with $\vec{h}_i^{(0)} = \vec{0}$) to produce *encoded inputs*, $\vec{z}_i^{(t)}$, as such:

$$\vec{z}_i^{(t)} = f_A(\vec{x}_i^{(t)}, \vec{h}_i^{(t-1)}) \tag{1}$$

The encoded inputs are then processed using the *processor network* $P$. The processor network shares its parameters among all algorithms being learnt. The processor network takes as input the encoded inputs $\mathbf{Z}^{(t)} = \{\vec{z}_i^t\}_{i \in V}$ and edge features $\mathbf{E}^{(t)} = \{\vec{e}_{ij}^{(t)}\}_{e \in E}$ and produces as output latent node features, $\mathbf{H}^{(t)} = \{\vec{h}_i^t \in \mathbb{R}^K\}_{i \in V}$:

$$\mathbf{H}^{(t)} = P(\mathbf{Z}^{(t)}, \mathbf{E}^{(t)}) \tag{2}$$

The node and algorithm specific outputs are then calculated by the *decoder-network*, $g_A$:

$$\vec{y}_i^{(t)} = g_A(\vec{z}_i^{(t)}, \vec{h}_i^{(t)}) \tag{3}$$

Note that the processor network also needs to make a decision on whether to terminate the algorithm. This is performed by an algorithm-specific *termination network*, $T_A$, which provides the probability of termination $\tau^{(t)}$—after applying the logistic sigmoid activation $\sigma$—as follows:

$$\tau^{(t)} = \sigma(T_A(\mathbf{H}^{(t)}, \overline{\mathbf{H}^{(t)}})) \tag{4}$$

where $\overline{\mathbf{H}^{(t)}} = \frac{1}{|V|} \sum_{i \in V} \vec{h}_i^{(t)}$ is the average node embedding. If the algorithm hasn't terminated (e.g. $\tau^{(t)} > 0.5$) the computation of Eqns. 1–4 is repeated—with parts of $\vec{y}_i^{(t)}$ potentially reused in $\vec{x}_i^{(t+1)}$.

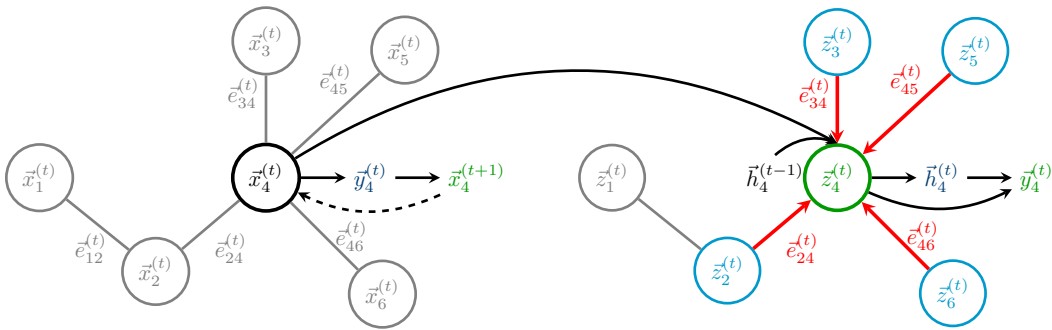

Figure 1: A visualisation of the relation between local computations of graph algorithms (**left**) and the neural graph algorithm executor (**right**). In graph algorithms, node values $\vec{y}_i^{(t)}$ (*e.g.* reachability, shortest-path distance) are updated at every step of execution. Analogously, the node values are predicted by the neural executor from the hidden representation $\vec{h}_i^{(t)}$ computed via message passing.

In our experiments, all algorithm-dependent networks, $f_A$, $g_A$ and $T_A$, are all linear projections, placing the majority of the representational power of our method in the processor network $P$. As we would like the processor network to be mindful of the structural properties of the input, we employ a graph neural network (GNN) layer capable of exploiting edge features as $P$. Specifically, we compare graph attention networks (GATs) (Veličković et al., 2018). (Equation 5, left) against message-passing neural networks (MPNNs) (Gilmer et al., 2017) (Equation 5, right):

$$\vec{h}_i^{(t)} = \text{ReLU}\left(\sum_{(j,i)\in E} a\left(\vec{z}_i^{(t)}, \vec{z}_j^{(t)}, \vec{e}_{ij}^{(t)}\right) \mathbf{W}\vec{z}_j^{(t)}\right) \qquad \vec{h}_i^{(t)} = U\left(\vec{z}_i^{(t)}, \bigoplus_{(j,i)\in E} M\left(\vec{z}_i^{(t)}, \vec{z}_j^{(t)}, \vec{e}_{ij}^{(t)}\right)\right)$$
(5)

where $\mathbf{W}$ is a learnable projection matrix, $a$ is an attention mechanism producing *scalar coefficients*, while $M$ and $U$ are neural networks producing *vector messages*. $\bigoplus$ is an elementwise aggregation operator, such as maximisation, summation or averaging. We use linear projections for $M$ and $U$.

Note that the processor network $P$ is *algorithm-agnostic*, and can hence be used to execute *several* algorithms simultaneously. Lastly, we note that the setup also easily allows for including edge-level outputs, and graph-level inputs and outputs—however, these were not required in our experiments.

## 3 EXPERIMENTAL SETUP

**Graph generation**    To provide our learner with a wide variety of input graph structure types, we follow prior work (You et al., 2018; 2019) and generate undirected graphs from seven categories:

- *Ladder* graphs;
- *2D grid* graphs;
- *Trees*, uniformly randomly generated from the Prüfer sequence;
- *Erdős-Rényi* (Erdős & Rényi, 1960) graphs, with edge probability $\min\left(\frac{\log_2 |V|}{|V|}, 0.5\right)$;
- *Barabási-Albert* (Albert & Barabási, 2002) graphs, attaching either four or five edges to every incoming node;
- *4-Community* graphs—first generating four disjoint Erdős-Rényi graphs with edge probability 0.7, followed by interconnecting their nodes with edge probability 0.01;
- *4-Caveman* (Watts, 1999) graphs, having each of their intra-clique edges removed with probability 0.7, followed by inserting $0.025|V|$ additional shortcut edges between cliques.

We additionally insert a self-edge to every node in the graphs, in order to support easier retention of self-information through message passing. Finally, we attach a real-valued weight to every edge, drawn uniformly from the range $[0.2, 1]$. These weight values serve as the sole edge features, $e_{ij}^{(t)}$, for

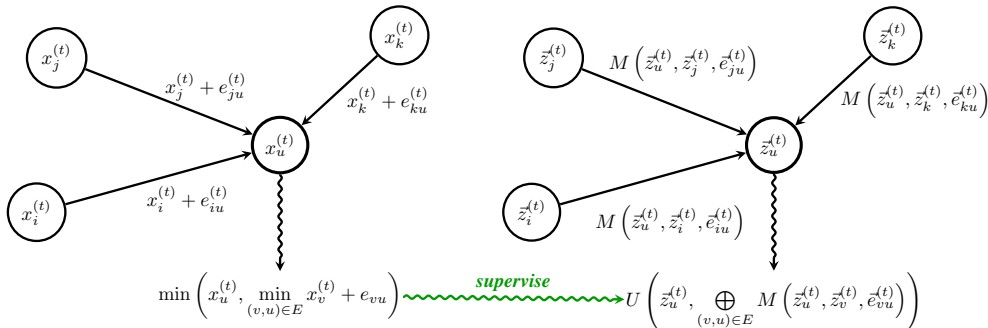

Figure 2: Illustrating the alignment of one step of the Bellman-Ford algorithm (**left**) with one step of a message passing neural network (**right**), and the supervision signal used for the algorithm learner.

all steps $t$. Note that sampling edge weights in this manner essentially guarantees the uniqueness of the recovered solution, simplifying downstream evaluation. We also ignore corner-case inputs (such as negative weight cycles), leaving their handling to future work.

We aim to study the algorithm execution task from a *"programmer"* perspective: human experts may manually inspect only relatively small graphs, and any algorithms derived from this should apply to arbitrarily large graphs. For each category, we generate 100 training and 5 validation graphs of only 20 nodes. For testing, 5 additional graphs of 20, 50 and 100 nodes are generated per category.

**Parallel algorithms** We consider two classical algorithms: *breadth-first search* for reachability, and the *Bellman-Ford* algorithm (Bellman, 1958) for shortest paths. The former maintains a single-bit value in each node, determining whether said node is reachable from a source node, and the latter maintains a scalar value in each node, representing its distance from the source node.

In both cases, the algorithm is initialised by randomly selecting the *source* node, $s$. As the initial input to the algorithms, $x_i^{(1)}$, we have:

$$\text{BFS}: x_i^{(1)} = \begin{cases} 1 & i = s \\ 0 & i \neq s \end{cases} \qquad \text{Bellman-Ford}: x_i^{(1)} = \begin{cases} 0 & i = s \\ +\infty & i \neq s \end{cases} \qquad (6)$$

This information is then propagated according to the chosen algorithm: a node becomes reachable from $s$ if any of its neighbours are reachable from $s$, and we may update the distance to a given node as the minimal way to reach any of its neighbours, then taking the connecting edge:

$$\text{BFS}: x_i^{(t+1)} = \begin{cases} 1 & x_i^{(t)} = 1 \\ 1 & \exists j. (j,i) \in E \wedge x_j^{(t)} = 1 \\ 0 & \text{otherwise} \end{cases} \qquad \text{B-F}: x_i^{(t+1)} = \min\left(\vec{x}_i^{(t)}, \min_{(j,i) \in E} x_j^{(t)} + e_{ji}^{(t)}\right)$$

$$(7)$$

For breadth-first search, no additional information is being computed, hence $y_i^{(t)} = x_i^{(t+1)}$. Additionally, at each step the Bellman-Ford algorithm may compute, for each node, the *"predecessor"* node, $p_i^{(t)}$ in the shortest path (indicating which edge should be taken to reach this node). This information is ultimately used to reconstruct shortest paths, and hence represents the crucial output:

$$\text{Bellman-Ford}: p_i^{(t)} = \begin{cases} i & i = s \\ \operatorname*{argmin}_{j;(j,i) \in E} x_j^{(t)} + e_{ji}^{(t)} & i \neq s \end{cases} \qquad (8)$$

Hence, for Bellman-Ford, $\vec{y}_i^{(t)} = p_i^{(t)} \| x_i^{(t+1)}$, where $\|$ is concatenation. To provide a numerically stable value for $+\infty$, we set all such entries to the length of the longest shortest path in the graph + 1.

We learn to execute these two algorithms *simultaneously*—at each step, concatenating the relevant $\vec{x}_i^{(t)}$ and $\vec{y}_i^{(t)}$ values for them. As both of the algorithms considered here (and most others) rely on

discrete decisions over neighbourhoods, learning to execute them should be naturally suited for the MPNN with the max-aggregator—a claim which we directly verify in the remainder of this section.

**Sequential algorithms** Unlike the previous two algorithms, single iterations of many classical graph algorithms will specifically focus on one node at a time—very often the case with *constructive* tasks. We seek to demonstrate that our neural graph algorithm execution paradigm aligns well with this setting too, and in this context we study *Prim's* algorithm (Prim, 1957) for minimum spanning trees.

Prim's algorithm maintains a partially constructed minimum spanning tree (MST)—initially, it is a singleton tree consisting of only a source node, $s$. At each step, Prim's algorithm searches for a new node to connect to the MST—chosen so that the edge attaching it to the tree is the *lightest* possible:

$$\text{Prim} : x_i^{(1)} = \begin{cases} 1 & i = s \\ 0 & i \neq s \end{cases} \qquad \text{Prim} : x_i^{(t+1)} = \begin{cases} 1 & x_i^{(t)} = 1 \\ 1 & \boxed{i = \underset{j \text{ s.t. } x_j^{(t)}=0}{\text{argmin}} \ \underset{k \text{ s.t. } x_k^{(t)}=1}{\min} e_{jk}^{(t)}} \\ 0 & \text{otherwise} \end{cases} \tag{9}$$

Once the new node is selected, the algorithm attaches it to the MST via this edge—similarly to Bellman-Ford, we can keep track of *predecessor nodes*, $p_i^{(t)}$:

$$\text{Prim} : p_i^{(t)} = \begin{cases} i & i = s \\ p_i^{(t-1)} & i \neq s \wedge x_i^{(t)} = 1 \\ \boxed{\underset{j \text{ s.t. } x_j^{(t)}=1}{\text{argmin}} \ e_{ij}^{(t)}} & x_i^{(t)} = 0 \wedge x_i^{(t+1)} = 1 \\ \bot \text{ (undefined)} & \text{otherwise} \end{cases} \tag{10}$$

In Equations 9–10, the boxed updates are the only modifications to the state of the algorithm at step $t$—centering only on the selected node to attach to the MST. Once again, the algorithm requires discrete decisions based on neighbourhood edge weights, hence we expect outperformance of MPNN-max.

For a visualisation of the expected alignment between a graph algorithm and our neural graph executors, refer to Figure 2. We provide an overview of all of the inputs and supervision signals used here for the executor in Appendix A.

**Neural network architectures** To assess the comparative benefits of different architectures for the neural algorithm execution task, we consider many candidate networks executing the computation of Equations 1–5, especially the processor network $P$: For the MPNN update rule, we consider maximisation, mean and summation aggregators. For the GAT update rule, we consider the originally proposed attention mechanism of Veličković et al. (2018), as well as Transformer attention (Vaswani et al., 2017); Additionally for GAT, we consider also attending over the *full graph*—adding a second attention head, only acting on the *non-edges* of the graph (and hence not accepting any edge features). The two heads' features are then concatenated and passed through another linear layer.

Analogously to our expectation that the best-performing MPNN rule will perform maximisation, we attempt to force the attentional coefficients of GAT to be as *sharp* as possible—applying either an *entropy penalty* to them (as in Ying et al. (2018)) or the *Gumbel softmax* trick (Jang et al., 2016).

We perform an additional sanity check to ensure that a GNN-like architecture is necessary in this case. Prior work (Xu et al., 2019) has already demonstrated the unsuitability of MLPs for reasoning tasks like these, and they will not support variable amounts of nodes. Here, instead, we consider an LSTM (Hochreiter & Schmidhuber, 1997) architecture into which serialised graphs are fed (we use an edge list, in a setup similar to (Graves et al., 2016)).

In all cases, the neural networks compute a latent dimension of $K = 32$ features, and are optimised using the Adam SGD optimiser (Kingma & Ba, 2014) on the binary cross-entropy for the reachability predictions, mean squared error for the distance predictions, categorical cross-entropy for the predecessor node predictions, and binary cross-entropy for predicting termination (all applied simultaneously). We use an initial learning rate of 0.0005, and perform early stopping on the validation accuracy for the predecessor node (with a patience of 10 epochs). If the termination network $T_A$ does not terminate the neural network computation within $|V|$ steps, it is assumed terminated at that point.

Table 1: Accuracy of predicting reachability at different test-set sizes, trained on graphs of 20 nodes. GAT* correspond to the best GAT setup as per Section 3 (GAT-full using the full graph).

| Model | Reachability (mean step accuracy / last-step accuracy) | | |
| --- | --- | --- | --- |
| | *20 nodes* | *50 nodes* | *100 nodes* |
| LSTM (Hochreiter & Schmidhuber, 1997) | 81.97% / 82.29% | 88.35% / 91.49% | 68.19% / 63.37% |
| GAT* (Veličković et al., 2018) | 93.28% / 99.86% | 93.97% / **100.0%** | 92.34% / **99.97%** |
| GAT-full* (Vaswani et al., 2017) | 78.40% / 77.86% | 85.76% / 91.83% | 88.98% / 91.51% |
| MPNN-mean (Gilmer et al., 2017) | **100.0% / 100.0%** | 61.05% / 57.89% | 27.17% / 21.40% |
| MPNN-sum (Gilmer et al., 2017) | 99.66% / **100.0%** | 94.25% / **100.0%** | 94.72% / 98.63% |
| MPNN-max (Gilmer et al., 2017) | **100.0% / 100.0%** | **100.0% / 100.0%** | **99.92%** / 99.80% |

Table 2: Accuracy of predicting the shortest-path predecessor node at different test-set sizes. (*curriculum*) corresponds to a curriculum wherein reachability is learnt first. (*no-reach*) corresponds to training without the reachability task. (*no-algo*) corresponds to the classical setup of directly training on the predecessor, without predicting any intermediate outputs or distances.

| Model | Predecessor (mean step accuracy / last-step accuracy) | | |
| --- | --- | --- | --- |
| | *20 nodes* | *50 nodes* | *100 nodes* |
| LSTM (Hochreiter & Schmidhuber, 1997) | 47.20% / 47.04% | 36.34% / 35.24% | 27.59% / 27.31% |
| GAT* (Veličković et al., 2018) | 64.77% / 60.37% | 52.20% / 49.71% | 47.23% / 44.90% |
| GAT-full* (Vaswani et al., 2017) | 67.31% / 63.99% | 50.54% / 48.51% | 43.12% / 41.80% |
| MPNN-mean (Gilmer et al., 2017) | 93.83% / 93.20% | 58.60% / 58.02% | 44.24% / 43.93% |
| MPNN-sum (Gilmer et al., 2017) | 82.46% / 80.49% | 54.78% / 52.06% | 37.97% / 37.32% |
| MPNN-max (Gilmer et al., 2017) | **97.13% / 96.84%** | **94.71% / 93.88%** | **90.91% / 88.79%** |
| MPNN-max (*curriculum*) | 95.88% / 95.54% | 91.00% / 88.74% | 84.18% / 83.16% |
| MPNN-max (*no-reach*) | 82.40% / 78.29% | 78.79% / 77.53% | 81.04% / 81.06% |
| MPNN-max (*no-algo*) | 78.97% / 95.56% | 83.82% / 85.87% | 79.77% / 78.84% |

For Prim's algorithm, as only one node at a time is updated, we optimise the categorical cross-entropy of predicting the next node, masked across all the nodes not added to the MST yet.

It should be noted that, when learning to execute Bellman-Ford and Prim's algorithms, the prediction of $p_i^{(t)}$ is performed by scoring each node-pair using an edge-wise scoring network (a neural network predicting a scalar score from $\vec{h}_i^{(t)}\|\vec{h}_j^{(t)}\|\vec{e}_{ij}^{(t)}$), followed by a softmax over all neighbours of $i$.

## 4 RESULTS AND DISCUSSION

**Parallel algorithm execution** In order to evaluate how faithfully the neural algorithm executor replicates the two parallel algorithms, we propose reporting the accuracy of predicting the reachability[2] (for breadth-first search; Table 1), as well as predicting the predecessor node (for Bellman-Ford; Table 2). We report this metric *averaged across all steps* $t$ (to give a sense of how well the algorithm is imitated across time), as well as the *last-step performance* (which corresponds to the final solution). While it is not necessary for recovering the final answer, we also provide the mean squared error of the models on the Bellman-Ford distance information, as well as the termination accuracy (computed at each step separately)—averaged across all timesteps—in Table 3.

The results confirm our hypotheses: the MPNN-max model exhibits superior generalisation performance on both reachability and shortest-path predecessor node prediction. Even when allowing for hardening the attention of GAT-like models (using entropy or Gumbel softmax), the more flexible

---

[2]Note that the BFS update rule is, in isolation, fully learnable by all considered GNN architectures. Results here demonstrate performance when learning to execute BFS simultaneously with Bellman-Ford. Specifically, in this regime, MPNN-sum's messages explode when generalising to larger sizes, while MPNN-mean dedicates most of its capacity to predicting the shortest path (cf. Tables 2–3).

Table 3: Mean squared error for predicting the intermediate distance information from Bellman-Ford, and accuracy of the termination network compared to the ground-truth algorithm, averaged across all timesteps. (*curriculum*) corresponds to a curriculum wherein reachability is learnt first. (*no-reach*) corresponds to training without the reachability task.

| Model | B-F mean squared error / mean termination accuracy | | |
| | *20 nodes* | *50 nodes* | *100 nodes* |
|---|---|---|---|
| LSTM (Hochreiter & Schmidhuber, 1997) | 3.857 / 83.43% | 11.92 / 86.74% | 74.36 / 83.55% |
| GAT* (Veličković et al., 2018) | 43.49 / 85.33% | 123.1 / 84.88% | 183.6 / 82.16% |
| GAT-full* (Vaswani et al., 2017) | 7.189 / 77.14% | 28.89 / 75.51% | 58.08 / 77.30% |
| MPNN-mean (Gilmer et al., 2017) | 0.021 / 98.57% | 23.73 / 89.29% | 91.58 / 86.81% |
| MPNN-sum (Gilmer et al., 2017) | 0.156 / 98.09% | 4.745 / 88.11% | $+\infty$ / 87.71% |
| MPNN-max (Gilmer et al., 2017) | **0.005** / 98.89% | **0.013 / 98.58%** | **0.238 / 97.82%** |
| MPNN-max (*curriculum*) | 0.021 / **98.99%** | 0.351 / 96.34% | 3.650 / 92.34% |
| MPNN-max (*no-reach*) | 0.452 / 80.18% | 2.512 / 91.77% | 2.628 / 85.22% |

Table 4: Shortest-path predecessor accuracy of the MPNN-max model trained jointly with the reachability objective on 20-node graphs, at different test graph sizes (up to 75× larger).

| Metric | MPNN-max predecessor prediction | | | | | |
| | *20 nodes* | *50 nodes* | *100 nodes* | *500 nodes* | *1000 nodes* | *1500 nodes* |
|---|---|---|---|---|---|---|
| Mean step accuracy | 97.13% | 94.71% | 90.91% | 83.08% | 77.53% | 74.90% |
| Last-step accuracy | 96.84% | 93.88% | 88.79% | 76.46% | 72.74% | 67.66% |

computational model of MPNN is capable of outperforming them. The performance gap on predicting the predecessor also widens significantly as the test graph size increases.

Our findings are compounded by observing the mean squared error metric on the intermediate result: with the MPNN-max being the only model providing a reasonable level of regression error at the 100-node generalisation level. It further accentuates that, even though models like the MPNN-sum model may also learn various thresholding functions—as demonstrated by (Xu et al., 2018)—aggregating messages in this way can lead to outputs of exploding magnitude, rendering the network hard to numerically control for larger graphs.

We perform two additional studies, executing the shortest-path prediction on MPNN-max *without predicting reachability*, and *without supervising on any intermediate algorithm computations*—that is, learning to predict predecessors (and termination behaviour) directly from the inputs, $x_i^{(1)}$. Note that this is the primary way such tasks have been tackled by graph neural networks in prior work. We report these results as *no-reach* and *no-algo* in Table 2, respectively.

Looking at the *no-reach* ablation, we observe clear signs of *positive knowledge transfer* occurring between the reachability and shortest-path tasks: when the shortest path algorithm is learned in isolation, the predictive power of MPNN-max drops significantly (while still outperforming many other approaches). In Appendix B, we provide a brief theoretical insight to justify this. Similarly, considering the *no-algo* experiment, we conclude that there is a clear benefit to supervising on the distance information—giving an additional performance improvement compared to the standard approach of only supervising on the final downstream outputs. Taken in conjunction, these two results provide encouragement for studying this particular learning setup.

Lastly, we report the performance of a *curriculum learning* (Bengio et al., 2009) strategy (as *curriculum*): here, BFS is learnt first in isolation (to perfect validation accuracy), followed by fine-tuning on Bellman-Ford. We find that this approach performs worse than learning both algorithms simultaneously, and as such we do not consider it in further experiments.

Desirable properties of the MPNN-max as an algorithm executor persist when generalising to even larger graphs, as we demonstrate in Table 4—demonstrating favourable generalisation on graphs up to 75× as large as the graphs originally trained on. We note that our observations also still hold

Table 5: The predictive performance of MPNN-max on 100-node graphs, after training on 20-node graphs of a particular type (Erdős-Rényi, or trees).

| | Reachability | | Predecessor | |
|---|---|---|---|---|
| **Graph type** | *From Erdős-Rényi* | *From trees* | *From Erdős-Rényi* | *From trees* |
| Ladder | 93.16% / 93.98% | 99.93% / 99.67% | 76.63% / 65.94% | 94.99% / 92.55% |
| 2-D Grid | 92.86% / 87.05% | 99.85% / 99.32% | 79.50% / 70.75% | 94.06% / 91.39% |
| Tree | 82.72% / 82.07% | 99.92% / 99.62% | 70.16% / 63.26% | 98.44% / 97.33% |
| Erdős-Rényi | 100.0% / 100.0% | 100.0% / 100.0% | 96.17% / 93.94% | 91.11% / 85.94% |
| Barabási-Albert | 100.0% / 100.0% | 100.0% / 100.0% | 94.91% / 92.90% | 83.90% / 75.79% |
| 4-Community | 100.0% / 100.0% | 100.0% / 100.0% | 90.01% / 86.38% | 75.88% / 64.04% |
| 4-Caveman | 100.0% / 100.0% | 100.0% / 100.0% | 91.55% / 90.04% | 80.02% / 72.06% |

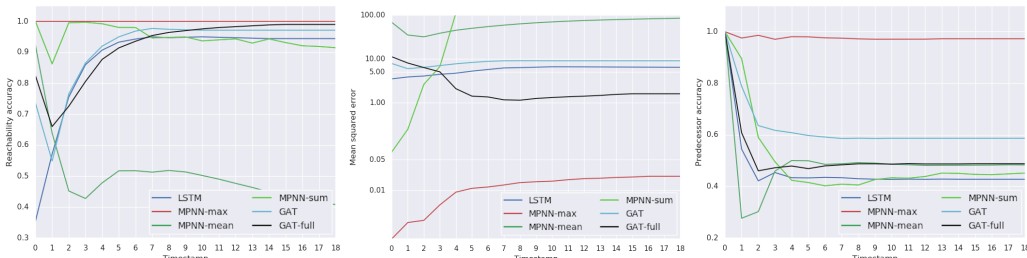

Figure 3: The per-step algorithm execution performances in terms of reachability accuracy (**left**), distance mean-squared error (**middle**) and predecessor accuracy (**right**), tested on 100-node graphs after training on 20-node graphs. Please mind the scale of the MSE plot.

when training on larger graphs (Appendix C). We also find that there is no significant overfitting to a particular input graph category—however we do provide an in-depth analysis of per-category performance in Appendix D.

**Additional metrics** The graphs we generate may be roughly partitioned into two types based on their *local regularity*—specifically, the ladder, grid and tree graphs all exhibit regular local structure, while the remaining four categories are more variable. As such, we hypothesise that learning from a graph of one such type only will exhibit better generalisation for graphs of the same type. We verify this claim in Table 5, where we train on either only Erdős-Rényi graphs or trees of 20 nodes, and report the generalisation performance on 100-node graphs across the seven categories. The results directly validate our claim, implying that the MPNN-max model is capable of biasing itself to the structural regularities found in the input graphs. Despite this bias, the model still achieves generalisation performances that outperform any other model, even when trained on the full dataset.

Further, we highlight that our choices of aggregation metrics may not be the most ideal way to assess performance of the algorithm executors: the last-step performance provides no indication of faithfulness to the original algorithm, while the mean-step performance may be artificially improved by terminating the algorithm at a latter point. While here we leave the problem of determining a better single-number metric to future work, we also decide to compound the results in Tables 1–2 by also plotting the test reachability/predecessor accuracies for each timestep of the algorithm individually (for 100-node graphs): refer to Figure 3.

Such visualisations can help identify cases where neural executors are "cheating", by e.g. immediately predicting every node is reachable: in these cases, we can see a characteristic—initially weak but steadily improving—performance curve. It also further solidifies the outperformance of MPNN-max.

Lastly, in Appendix E we apply the recently proposed GNNExplainer (Ying et al., 2019) model to detecting which graph substructures contributed the most to certain predictions.

**Sequential algorithm execution** We demonstrate results for all considered architectures on executing Prim's algorithm within Table 6. We provide the accuracy of predicting the next MST node

Table 6: Accuracy of selecting the next node to add to the minimum spanning tree, and predicting the minimum spanning tree predecessor node—at different test-set sizes. (*no-algo*) corresponds to the classical setup of directly training on the predecessor, without adding nodes sequentially.

| | Accuracy (next MST node / MST predecessor) | | |
|---|---|---|---|
| **Model** | *20 nodes* | *50 nodes* | *100 nodes* |
| LSTM (Hochreiter & Schmidhuber, 1997) | 11.29% / 52.81% | 3.54% / 47.74% | 2.66% / 40.89% |
| GAT* (Veličković et al., 2018) | 27.94% / 61.74% | 22.11% / 58.66% | 10.97% / 53.80% |
| GAT-full* (Vaswani et al., 2017) | 29.94% / 64.27% | 18.91% / 53.34% | 14.83% / 51.49% |
| MPNN-mean (Gilmer et al., 2017) | **90.56% / 93.63%** | 52.23% / 88.97% | 20.63% / 80.50% |
| MPNN-sum (Gilmer et al., 2017) | 48.05% / 77.41% | 24.40% / 61.83% | 31.60% / 43.98% |
| MPNN-max (Gilmer et al., 2017) | 87.85% / 93.23% | **63.89% / 91.14%** | **41.37% / 90.02%** |
| MPNN-max (*no-algo*) | — / 71.02% | — / 49.83% | — / 23.61% |

(computed against the algorithm's "ground-truth" ordering), as well as the accuracy of reconstructing the final MST (via the predecessors).

As anticipated, our results once again show strong generalisation outperformance of MPNN-max. We additionally compared against a non-sequential version (*no-algo*), where the MPNN-max model was trained to directly predict predecessors (without requiring sequentially chosing nodes). This resulted in poor generalisation to larger graphs, weaker than even the LSTM sequential baseline.

The insights from our setup verify that our neural graph execution paradigm is applicable to sequential algorithm execution as well—substantially expanding its range of possible applications.

## 5    CONCLUSIONS

In this manuscript, we have presented the **neural graph algorithm execution** task, where—unlike prior approaches—we optimise neural networks to imitate individual steps and all intermediate outputs of classical graph algorithms, parallel as well as sequential. Through extensive evaluation—especially on the tasks of reachability, shortest paths and minimum spanning trees—we have determined a highly suitable architecture in maximisation-based message passing neural networks, and identified clear benefits for multi-task learning and positive transfer, as many classical algorithms share related subroutines. We believe that the results presented here should serve as strong motivation for further work in the area, attempting to learn more algorithms simultaneously and exploiting the similarities between their respective subroutines whenever appropriate.

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

Table 7: Summary of inputs and supervision signals of the three algorithms considered.

| Algorithm | Inputs | Supervision signals |
|---|---|---|
| Breadth-first search | $x_i^{(t)}$: is $i$ reachable from $s$ in $\leq t$ hops? | $x_i^{(t+1)}$, 
 $\tau^{(t)}$: has the algorithm terminated? |
| Bellman-Ford | $x_i^{(t)}$: shortest distance from $s$ to $i$ 
 (using $\leq t$ hops) | $x_i^{(t+1)}$, 
 $\tau^{(t)}$, 
 $p_i^{(t)}$: predecessor of $i$ in the 
 shortest path tree (in $\leq t$ hops) |
| Prim's algorithm | $x_i^{(t)}$: is node $i$ in the (partial) MST 
 (built from $s$ after $t$ steps)? | $x_i^{(t+1)}$, 
 $\tau^{(t)}$, 
 $p_i^{(t)}$: predecessor of $i$ in the partial MST |

## A  SUMMARY OF ALGORITHM INPUTS AND SUPERVISION SIGNALS

To aid clarity, within Table 7, we provide an overview of all the inputs and outputs (supervision signals) for the three algorithms considered here (breadth-first search, Bellman-Ford and Prim).

## B  THEORETICAL INSIGHTS

We provide a brief theoretical insight into why learning to imitate multiple algorithms simultaneously may provide benefits to downstream predictive power.

Our insight comes from an information-theoretic perspective. Consider two algorithms, $A$ and $B$, that both operate on the same input, $x$, and produce outputs $y_A$ and $y_B$, respectively. We consider the task of learning to execute $A$ (that is, predicting $y_A$ from $x$), with and without $y_B$ provided[3]. We operate on the further assumption that $A$ and $B$ *share subroutines*—implying that, knowing $x$, there is information content preserved between $y_A$ and $y_B$. Formally, we say that the *conditional mutual information* of the corresponding random variables, $I(Y_A; Y_B | X)$, is *positive*.

Expanding out the expression for $I(Y_A; Y_B | X)$, denoting Shannon entropy by $H$, we obtain:

$$\begin{aligned} I(Y_A; Y_B | X) &= H(Y_A | X) + H(Y_B | X) - H(Y_A, Y_B | X) \\ &= H(Y_A | X) + H(Y_B | X) - (H(Y_A | Y_B, X) + H(Y_B | X)) \\ &= H(Y_A | X) - H(Y_A | Y_B, X) \end{aligned} \tag{11}$$

As $I(Y_A; Y_B | X) > 0$, we conclude $H(Y_A | X) > H(Y_A | Y_B, X)$; therefore, providing $y_B$ upfront strictly reduces the information-theoretic uncertainty in $y_A$, thus making it potentially more suitable for being learned by optimisation techniques.

## C  LARGER-SCALE STUDIES

To investigate the models' behaviour when generalising to larger graphs, we conduct experiments for executing the two parallel algorithms (BFS and Bellman-Ford) when training on graphs with 100 nodes, and testing on graphs with 1000 nodes, as reported in Table 8. These results further solidify the outperformance of MPNN-based models, even outside of the studied "programmer" regime.

## D  PERFORMANCE PER GRAPH TYPE

As our training and testing graphs come from a specific set of seven categories, it is natural to study the predictive power of the model conditioned on the testing category. Firstly, in Table 9, we provide

---

[3]Note that here we're implicitly assuming that $y_B$ is trivial enough to be fully learnt on its own—and thus can be provided to the model. This is a more strict way of assuming that $B$ is a "simpler" algorithm than $A$.

Table 8: Results of scaling to large graphs with 1000 nodes, while training on graphs with 100 nodes.

| Model | Reachability | Predecessor |
|---|---|---|
| LSTM | 66.63% / 72.62% | 33.73% / 32.36% |
| GAT* | 83.43% / 89.15% | 37.53% / 36.16% |
| MPNN-max | **100.0% / 99.99%** | **96.45% / 96.25%** |

Table 9: The predictive performance of MPNN-max on 100-node graphs—after training on 20-node graphs—partitioned by graph type.

| Graph type | Reachability | Predecessor |
|---|---|---|
| Ladder | 95.57% / 98.63% | 94.13% / 91.47% |
| 2-D grid | 95.93% / 93.28% | 87.90% / 83.77% |
| Tree | 99.55% / 98.32% | 98.60% / 97.83% |
| Erdős-Rényi | 100.0% / 100.0% | 94.00% / 89.65% |
| Barabási-Albert | 100.0% / 100.0% | 92.71% / 88.60% |
| 4-Community | 100.0% / 100.0% | 86.25% / 79.65% |
| 4-Caveman | 100.0% / 100.0% | 91.55% / 86.96% |

the per-category results of predicting reachability and predecessor for MPNN-max. We report that the performance is roughly evenly distributed across the categories—with trees being the easiest to learn on and grids/community graphs the hardest. These results align well with our expectations:

- In trees, computing the shortest path tree is equivalent to a much simpler task—*rooting* the input tree in the source vertex—that a model may readily pick up on.

- Making proper choices on grids requires propagating decisions over *long trajectories*[4]—as such, a poor decision early on may more strongly compromise the overall performance of retrieving the shortest path tree.

- As the community graphs are composed of four interconnected *dense* graphs (Erdős-Rényi with $p = 0.7$), the *node degree distribution* the model is required to handle may change drastically as graphs increase in size. This may require the model to aggregate messages over substantially larger neighbourhoods than it is used to during training.

## E   EXPLAINING GNN PREDICTIONS

We provide a further qualitative analysis of what the MPNN-max architecture has learnt when performing algorithm execution. In particular, we apply a model similar to GNNExplainer (Ying et al., 2019) for explaining the decisions made during the neural execution. For the reachability task, the explainer asnwers the question: *"for a given node $u$, which node in the neighbourhood of $u$ influences the reachability prediction made by the neural execution model?"*.

We use the best performing model, MPNN-max, to demonstrate the explanation. Given an already trained model on graphs with 20 nodes, starting from any node $u$ of the neural execution sequence, we optimise for an adjacency mask $\mathbf{M}$, that is initialized to 1 for all edges that connect to $u$, and 0 everywhere else. Instead of the original adjacency matrix $\mathbf{A}$, we use $\mathbf{A} \odot \sigma(\mathbf{M})$ as the input adjacency to the model. We fix the model parameters and only train on the adjacency mask using the same reachability loss, with the additional term of the sum of values in the adjacency mask. This encourages the explanation to remove as many edges as possible from the immediate neighbourhood of $u$, while still being able to perform the correct reachability updates.

When the mask is trained until convergence, we pick the edge that has the maximum weight in the mask to be the predecessor that explains the reachability of the node.

---

[4]Note that this is also the case with ladder graphs, but these trajectories cannot get very complicated in this case, as the ladder graph is a product of two path graphs.

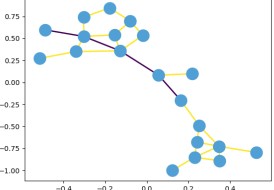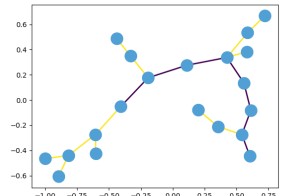

Figure 4: Identified reachability paths for a noisy Caveman graph and tree graph, using GNNExplainer. The purple edges indicate the predecessor relationships identified by the explainer, while the yellow edges are the remainder of the graph's edges.

We then perform the same explanation procedure on the ground-truth predecessor of $u$ in the BFS algorithm, and continue the process until we reach the source node, $s$ of the BFS algorithm. If all explanations are correct, we will observe a path that connects the node $u$ to the starting node of the BFS algorithm. Any disconnection indicates an incorrect explanation that deviates from the ground-truth, which could either be due to the incorrect prediction of the model, or an incorrect explanation. Using the standard training dataset in our experiments, we observe that $82.16\%$ of the instances have a path explanation with *no error* in the explanation compared to the ground-truth predecessors. Two of these examples are visualised in Figure 4. Additionally, $93.85\%$ of the predecessor explanation correspond to the ground-truth, providing further qualitative insight into the algorithm execution capacity of MPNN-max.

