# OpenReview forum: "Neural Execution of Graph Algorithms"
_ICLR.cc/2020/Conference — Accept (Poster)_

### Official Review · AnonReviewer3 · 2019-10-22
**Official Blind Review #3**

**Rating:** 8

**Review:**

The paper suggests training neural network to imitate graph algorithms in a more fine-grained way than done before: by learning primitives and subroutines rather than the final output. The paper makes quite a strong case for the advantage of this approach, citing the fact that many graph algorithms share subroutines, which could simplify learning and support joint training and transfer learning. The experiments are detailed an elaborate.

The main weakness I see is the size of the graphs in the experiments. They are mainly limited to graphs with up to 100 nodes, with additional brief results for 1000 nodes in the appendix. These sizes are so small so as to raise doubts if the reported accuracy results would indeed scale, or whether they might require a significantly heavier network architecture. Moreover graphs of this size do not actually pose any difficulty for classic graph algorithms, that would justify invoking such heavy cannons like neural networks.

Nonetheless, I find this to be a conceptually strong paper with interesting ideas and thorough experiments (which in the least establish proof of concept; I am willing to accept leaving the issue of handling larger graphs to future work). I think the paper should be accepted.

Other comments:
1. The legend font in Figure 3 is to small (I am positively unable to read it off page) and the MPNN-sum plot is invisible in print. I hope the authors can reproduce the plots more clearly.
2. I don't quite see the point in Appendix A, brief as it is. It apparently just states the fact that if two random variables share mutual information then knowing one reduces the entropy of the other. This is rather obvious both intuitively and formally.

**Experience Assessment:**

I have read many papers in this area.

**Review Assessment: Checking Correctness Of Derivations And Theory:**

I carefully checked the derivations and theory.

**Review Assessment: Checking Correctness Of Experiments:**

I assessed the sensibility of the experiments.

**Review Assessment: Thoroughness In Paper Reading:**

I read the paper at least twice and used my best judgement in assessing the paper.

---

> ### Author Response · Authors · 2019-11-15
> **Reply to AnonReviewer3**
>
> Thank you very much for the kind review, and we are very glad you enjoyed the paper!
>
> As recommended, we have now provided additional experiments, testing the MPNN-max model on graphs of 500, 1000 and 1500 nodes (after training on graphs of 20 nodes). They may be found in Table 4. We find that the generalisation properties of MPNN-max carry over to these graph sizes and further solidify our contribution in terms of scale. We agree that graphs of these sizes don’t really pose any problems for classical algorithms, and instead focus on the multi-task and transfer learning aspect, which could enable us to tackle more challenging problems in the future.
>
> We have also increased the size of the plots and modified the colour of the MPNN-sum curve in Figure 3, hoping that this will further aid clarity.

---

### Official Review · AnonReviewer1 · 2019-10-23
**Official Blind Review #1**

**Rating:** 8

**Review:**

This paper investigates using GNNs to learn graph algorithms. It proposes a model which consists of algorithm-dependent encoder and decoder, and algorithm-independent processor.
Authors try to learn BFS, Bellman-Ford and Prim algorithms on various types of random graphs.
Experimental results suggest that MPNN with max aggregator outperforms other variants significantly in terms of generalization.

Overall, this paper presents a solid contribution on learning graph algorithms using GNNs, despite the caveat for some clarifications on the model and experiments.
Given these clarifications in an author response, I would be willing to increase the score.

Pros:

1, Although it is less satisfying to learn to solve graph problems where polynomial-time algorithms exist, I still appreciate the contribution of the paper, especially the algorithm-independent processor. It is one step forward to models which could learn meta-level representation of algorithms. The findings of this paper may suggest that different types of operators used in the GNN may have different inductive bias in learning different types of algorithms.

2, Experimental comparisons are adequate and convincing. The detailed analysis of empirical results also provide good explanation.

Cons & Suggestions:

1, Given the dynamic programming nature of the most of tasks, it is not that surprising that MPNN with max aggregation could solve them pretty well. What surprises me is that MPNN-mean / MPNN-sum could not generalize well (i.e., performance drops significantly on 50 and 100 nodes settings) on the reachability task. In my opinion, the reachability task could be easily handled by any diffusion/propagation based models including, e.g., MPNN, GCN, GAT, as long as the information is spread over the input graph. Could you explain why does this happen?

2, Training details are sparse. If I understood correctly, the training of various GNNs is done by teacher forcing such that each step some supervised information collected from the underlying graph algorithms is provided to GNNs. However, it is not clear that what supervised information is exactly provided under each graph algorithm. It would be great to have a table to summarize what the input and supervised output information is for each graph algorithm.

3, The processor is trained to learn multiple graph algorithms simultaneously. A natural question to ask is how does the performance change when you train the processor with different combinations of algorithms? What is the impact of correlations between different graph algorithms? Did you explore learning with graph algorithms following some sequential schedule, e.g., curriculum learning?

4, It would be great to provide more results on larger size graphs, e.g., trained on 20 nodes but test on 1000 or more nodes. I saw one set of experiments trained on 100 and test 1000 nodes in the appendix. More results along this line would make the paper more convincing on the generalization.

5, It would be great to discuss [1] as it also studies how to use GNNs to learn a special class of graph algorithms, i.e., probabilistic inference algorithms on graphs. It is shown in [1] that belief propagation algorithm could be seen as a specially constructed GNN. Interestingly, some well-known graph algorithms could be re-formulated as probabilistic inference algorithms, e.g., graph-cut could be reformulated as max-product [2].

[1] Yoon, K., Liao, R., Xiong, Y., Zhang, L., Fetaya, E., Urtasun, R., Zemel, R. and Pitkow, X., 2018. Inference in probabilistic graphical models by graph neural networks. arXiv preprint arXiv:1803.07710.
[2] Tarlow, D., Givoni, I.E., Zemel, R.S. and Frey, B.J., 2011, July. Graph Cuts is a Max-Product Algorithm. In UAI (pp. 671-680).

======================================================================================================

Thanks for the thorough response! It resolves my concerns. I improved my score.


**Experience Assessment:**

I have published in this field for several years.

**Review Assessment: Checking Correctness Of Derivations And Theory:**

I carefully checked the derivations and theory.

**Review Assessment: Checking Correctness Of Experiments:**

I assessed the sensibility of the experiments.

**Review Assessment: Thoroughness In Paper Reading:**

I read the paper at least twice and used my best judgement in assessing the paper.

---

> ### Author Response · Authors · 2019-11-15
> **Reply to AnonReviewer1**
>
> Thank you for the very careful review and kind words about our contributions.
>
> We provide responses to your comments in turn, hoping that they have adequately addressed your concerns. We are happy to discuss further on any of these points, of course!
>
> 1. Regarding the performance of MPNN-mean/sum on reachability, it should be highlighted that the results are presented in the *multi-task* setup, where reachability has to be learnt simultaneously with shortest-paths. In this case, the summation architecture suffers from exploding messages (exemplified by the NaN values on MSE), while the averaging architecture ends up allocating most of its capacity to learning Bellman-Ford. We confirm that, when learnt in isolation, all of these architectures are capable of strongly generalising on reachability, and have now made that point clear in the paper (see Footnote 2 on page 6).
>
> 2. Thank you for the comment regarding the clarity of the experimental setup! We fully agree, and now provide a table summarising all the inputs and supervision signals used -- see Table 7 in Appendix A. We also refer to this Appendix properly within Section 3.
>
> 3. Initially, we haven’t experimented with a sequential schedule. As per your recommendation, we provide results for a “curriculum” strategy, in which a network is first pre-trained on breadth-first search (until reaching perfect validation accuracy), followed by learning to imitate Bellman-Ford. These results are added in Tables 2 and 3 as an additional row, and demonstrate that such a sequential learning strategy performs worse than the simultaneous one in this case -- although it still provides benefits over the no-algo variant.
>
> 4. As recommended, we have now provided additional experiments, testing the MPNN-max model on graphs of 500, 1000 and 1500 nodes (after training on graphs of 20 nodes). They may be found in Table 4. We find that the generalisation properties of MPNN-max carry over to these graph sizes and further solidify our contribution in terms of scale.
>
> 5. We now directly cite reference [1] in the paper -- namely, in Section 1, where we mention other related work. Thank you for pointing this reference out to us, it is certainly relevant to our work!

---

### Official Review · AnonReviewer2 · 2019-10-25
**Official Blind Review #2**

**Rating:** 1

**Review:**

Summary:

This paper uses message-passing neural networks to learn to predict which
nodes to update with what metadata. Experiments are shown that the algorithm
can learn to visit nodes in the same order as a breadth-first-search as well
as the Bellman-Ford shortest-path algorithm.

Feedback:

I'm not sure what problem this paper is trying to solve. We are provided
the graph and graph algorithm as input to the network so that we can
learn what the algorithm will do next? I'm not sure why this is a problem?
Are these algorithms badly written and need to be improved? Do we want to
learn how to execute graph algorithms in general?

I found the paper fairly hard to read and follow. I wish the
model were described all in one place and the related work also
in just one place. I think this paper would be greatly improved
by more work on explaining the motivation as well as more
clearly. I also feel the paper could be much stronger if it was grounded
in an existing problem others in the community might have.

Clarifications:

In page 4, I don't know what it means to execute these two algorithms
simulatenously?

**Experience Assessment:**

I do not know much about this area.

**Review Assessment: Checking Correctness Of Derivations And Theory:**

N/A

**Review Assessment: Checking Correctness Of Experiments:**

I did not assess the experiments.

**Review Assessment: Thoroughness In Paper Reading:**

I read the paper at least twice and used my best judgement in assessing the paper.

---

> ### Author Response · Authors · 2019-11-15
> **Response to AnonReviewer2**
>
> We thank the reviewer for their comments. We would like to address a number of points raised here:
>
> 1. As pointed out by the review, we trained a graph neural network to execute a graph algorithm on an arbitrary graph. Our objective is to enforce an inductive bias towards “algorithmic reasoning” within the GNN’s update rule. This bias can be a very useful prior for e.g. discovering novel algorithms, or improving existing ones. We also experimentally demonstrate transfer between learning two different algorithms (BFS and Bellman-Ford), which directly points out the potential for one learnt algorithm executor to reinforce the predictions of another.
>
> 2. Besides the transfer learning aspect, the algorithm execution task is a challenging problem in its own right, as shown in our baseline experiments (namely, many state-of-the-art GNN architectures such as GATs or MPNN variants are unable to generalise). Note that previous work has studied specific cases of these problems as well (e.g., Pointer Networks, Neural Turing Machines, etc).
>
> 3. In the current version of the paper, our model is explained in one place (in Section 2), and our related work is outlined in one place (in the Introduction). Is there a specific part of the submission the reviewer would like us to move to the model description to help improve clarity of the paper?
>
> 4. We have upgraded the clarity of the paper now by making substantial changes and performing additional experiments, such as:
>
> * Summarising all inputs and supervised signals of the algorithms considered here (Table 7 in Appendix A);
> * Additional related work cited throughout the paper;
> * Ablation studies, such as testing the generalisation of MPNN-max on larger graphs and assessing the benefits of a “curriculum learning” strategy.
>
> 5. Regarding executing two algorithms simultaneously, this is done by, at each step, concatenating the relevant x(t) and y(t) together as inputs/targets for the network. We mention this at the bottom of page 4, right after introducing the term “simultaneously”.
>
> We are happy to discuss further on any of these points!

---

### Author Response · Authors · 2019-11-15
**Summary of revisions made to the paper in the discussion period**

We hope that the revisions we have made to the paper have properly addressed the comments of the reviewers on our work - and that its overall contribution, quality and clarity is now improved! We would like to thank everyone once again for their thoughtful comments on our paper.

We provide a summary of the changes made to the paper:

* We have now provided additional experiments, testing the MPNN-max model on graphs of 500, 1000 and 1500 nodes (after training on graphs of 20 nodes). They may be found in Table 4. We find that the generalisation properties of MPNN-max carry over to these graph sizes and further solidify our contribution in terms of scale.

*  We provide results for a “curriculum learning” strategy, in which a network is first pre-trained on breadth-first search (until reaching perfect validation accuracy), followed by learning to imitate Bellman-Ford. These results are added in Tables 2 and 3 as an additional row, and demonstrate that such a sequential learning strategy performs worse than the simultaneous one in this case -- although it still provides benefits over the no-algo variant.

* To aid clarity, we summarise all inputs and supervised signals of the algorithms considered here, within Table 7 in Appendix A;

* We confirm that, when learnt in isolation, all of the GNN architectures considered are capable of strongly generalising on the reachability task, and have now made that point clear in the paper (see Footnote 2 on page 6).

* In Section 1, we now cite the related work of Yoon et al.

* We have increased the size of the plots and modified the colour of the MPNN-sum curve in Figure 3, hoping that this will further aid clarity.

---

### Public Comment · ~Le_Song1 · 2020-02-19
**Interesting paper**

This an interesting paper on representing/learning graph algorithms with graph neural networks.
We have two very related papers on this direction:
1. Learning Combinatorial Optimization Algorithms over Graphs. Hanjun Dai, Elias B. Khalil, Yuyu Zhang, Bistra Dilkina, Le Song. NeurIPS 2017.
2. Learning Steady-States of Iterative Algorithms over Graphs. Hanjun Dai, Zornitsa Kozareva, Bo Dai, Alexander J. Smola, Le Song. ICML 2018.
Discussing them in the paper can enrich the context of the paper.

---

### Decision · Program_Chairs · 2019-12-19

**Decision:**

Accept (Poster)

**Comment:**

It seems to be an interesting contribution to the area. I suggest acceptance.